# Climate Warming Does Not Override Eutrophication, but Facilitates Nutrient Release from Sediment and Motivates Eutrophic Process

**DOI:** 10.3390/microorganisms11040910

**Published:** 2023-03-31

**Authors:** Huan Wang, Qi Li, Jun Xu

**Affiliations:** 1State Key Laboratory of Marine Resource Utilization in South China Sea, Hainan University, Haikou 570228, China; 2Institute of Hydrobiology, Chinese Academy of Sciences, Wuhan 430072, China

**Keywords:** warming, eutrophication, mesocosm, nutrient fluxes, shallow lake

## Abstract

The climate is changing. The average temperature in Wuhan, China, is forecast to increase by at least 4.5 °C over the next century. Shallow lakes are important components of the biosphere, but they are sensitive to climate change and nutrient pollution. We hypothesized that nutrient concentration is the key determinant of nutrient fluxes at the water-sediment interface, and that increased temperature increases nutrient movement to the water column because warming stimulates shifts in microbial composition and function. Here, twenty-four mesocosms, mimicking shallow lake ecosystems, were used to study the effects of warming by 4.5 °C above ambient temperature at two levels of nutrients relevant to current degrees of lake eutrophication levels. This study lasted for 7 months (April–October) under conditions of near-natural light. Intact sediments from two different trophic lakes (hypertrophic and mesotrophic) were used, separately. Environmental factors and bacterial community compositions of overlying water and sediment were measured at monthly intervals (including nutrient fluxes, chlorophyll *a* [chl *a*], water conductivity, pH, sediment characteristics, and sediment-water et al.). In low nutrient treatment, warming significantly increased chl *a* in the overlying waters and bottom water conductivity, it also drives a shift in microbial functional composition towards more conducive sediment carbon and nitrogen emissions. In addition, summer warming significantly accelerates the release of inorganic nutrients from the sediment, to which microorganisms make an important contribution. In high nutrient treatment, by contrast, the chl *a* was significantly decreased by warming, and the nutrient fluxes of sediment were significantly enhanced, warming had considerably smaller effects on benthic nutrient fluxes. Our results suggest that the process of eutrophication could be significantly accelerated in current projections of global warming, especially in shallow unstratified clear-water lakes dominated by macrophytes.

## 1. Introduction

Shallow lakes dominate inland waters, where eutrophication due to human activities and seasonal deviations in temperature and hydrology due to climate change are considered two major threats to shallow aquatic ecosystems worldwide [1,2,3]. Their large surface-to-volume ratio makes them vulnerable to environmental changes, thus shifting between a state of high biodiversity and plant dominance and a state shaded by suspended sediments or phytoplankton [4]. Increased nutrient fluxes in shallow lake ecosystems influenced by temperature increases induced by global climate change [5,6] changed the structure and function of shallow lakes [3]. Sediment, as a source and sink for nutrients in overlying water [7,8], determines the dynamics of the income and expenditure cycle dynamics and sustains the primary productivity of aquatic ecosystems [9,10]. There is a lack of knowledge on how climate change affects nutrient dynamics and fluxes in lake ecosystems [5,11].

Under eutrophic conditions, elevated temperature increases nutrient release from lake sediments, and even after external nutrient loading is reduced, nutrients transferred from the sediment to the water column become the main source of nutrients, which prevents water quality improvement for a considerable period of time [12,13,14,15]. Biogeochemical factors, including temperature, light, pH, dissolved oxygen, organic matter content, redox conditions, and microbial-related decomposition processes all influence nutrient cycling of the water column sediment interface [15,16,17,18,19]. It is believed that among these several physical, chemical, and biological factors, elevated temperature is the primary cause of nutrient fluxes in sediments [15,20,21]. This is demonstrated by the fact that higher temperatures alter the biogeochemical processes in the lake and have an impact on the nutrient balance cycling and flux rates by decreasing oxygen solubility [15,22], increasing the mineralization of organic matter, and accelerating inorganic sorption and precipitation rates [20,23]. In earlier research, different temperatures were used to examine variations in sediment nutrient fluxes and transformations in rivers, lakes, and reservoirs [13,17], but it is still less clear how warming is affecting nutrient transformations and fate.

Warming promotes enhanced microbial metabolism at the water-sediment interface, which may increase the release of inorganic compounds and the decomposition of organic matter, reduce sediment oxygen concentrations, and result in a decrease in sediment redox potential [24,25]. For freshwater ecosystems, sediments play a crucial role as both a source and a sink of nutrients, [26], enriching nutrients through abiotic sorption, biological uptake [27,28], or release back into the water column through microbial decomposition or reduction [24]. High temperatures induce carbon release [15,16] and inhibit nitrification, but promoting denitrification to release NH4^+^-N [22,29] toward mineralization of organic nutrients by enhancing microbial metabolism in the sediment [14,15,23]. Thus, global warming may accelerate eutrophication in aquatic ecosystems by altering the composition and function of microbial communities and changing the direction and intensity of nutrient transfer processes at the sediment-water interface [20,30].

According to the IPCC (the Intergovernmental Panel on Climate Change) fifth assessment report (AR5), surface temperatures in South Asia are anticipated to rise by 3–5 °C during 2081~2100 [31]. Given how sensitive shallow lakes are, the future water temperature is anticipated to have a significant impact on aquatic ecosystems in the future [32,33]. The impact of temperature on the dynamics of shallow lakes at different nutrient levels and their nutrient transformations remain unclear. Here, we conducted an experiment aimed at stimulating the prediction of possible future temperatures, and we used 24 temperature-controlled 2.5 m^3^ intermediate test chambers divided into two thermal states (ambient and +4.5 °C) and two nutrient level states. The major goal of this study was to determine whether sediments in shallow lakes at various nutrient levels act as a source or sink of nutrients during global warming [34]. We tested three hypotheses: (1) high water temperatures will accelerate sediment C and N transformations and fluxes due to organic matter decomposition; (2) eutrophication processes may be significantly accelerated in current global warming projections, particularly in shallow unstratified clear-water lakes dominated by macrophytes; (3) The transformation of the source and sink of nutrient fluxes at the sediment and water interface is largely dependent on the coupling of microbial community structure and function in sediment and water, which responds to climate warming.

## 2. Materials and Methods

### 2.1. Mesocosm Establishment

Our mesocosm system was located at Wuhan City in the center of China (30°28′ N, 114°21′ E) and included 24 tanks, each 1.5 m deep and 1.5 m in diameter. The mesocosm was circular polyethylene tanks containing 1760 L of water and sediment. The heating and control treatments were randomly designed to be distributed, with the control system being the seasonal temperature in its natural state and the heating treatments were heated using aquarium heaters (600 W), plus 4.5 °C. The computer-controlled temperature control system regulated each temperature rise and fall individually according to the different treatments in combination with a temperature sensor at a depth of 30 cm (measured every 0.1 s). All treatments use a small pump to circulate the water column. 

In March, we collected sediments and water column samples from two contrasting nutrient level lakes located in the middle reaches of Yangtze River in China, a nutrient-rich plankton-dominated lake (TN and TP concentration were 3.85 and 1.50 mg/g in sediments, 2.250 and 0.198 mg/L in the water column, N 30°29′28″, E 114°21′34″) and a less nutrient-rich with macrophytes lake (TN and TP concentration were 1.53 and 0.41 mg/g in sediments, 0.432 and 0.023 mg/L in the water column, N 30°11′3″, E 114°37′59″). Using a Peterson grab sampler and a pump, the top few centimeters of sediment (0–5 cm) and the water column were collected in the pelagic zone of the lakes. After being sufficiently mixed separately, the samples were transferred into pre-cleaned containers and transported to the experimental mesocosms.

One meter of lake water and 10 cm of mixed lake sediment were added to each experimental mesocosm. The nutrient-rich sediment and its water originated from a nutrient-rich lake, while for the treatment of low nutrient level sediment derived from a less nutrient-rich lake. Before adding water to the mesocosm system, a 20-μm plankton net was employed to remove big vegetation detritus and prevent the uncontrolled ingress of macrovertebrates (e.g., fish or amphibians). It is worth mentioning that our control system does not include fish, as fish predation may lead to considerable changes in crustacean biomass and obscure the effects of warming and eutrophication. The sediments contain all native species and trophic levels in addition to their inherent complexity and variability.

There were four different treatments: a 4.5 °C temperature increase compared to the high nutrient (HNT), a high nutrient (HN) that mimicked the concurrent state in a high nutrient level lake, a low nutrient (LN) that mimicked the concurrent state in a low nutrient level lake, and a low nutrient with a 4.5 °C temperature increase (LNT). Each of the four treatments was produced in six copies. Four treatments were dispersed at random over a 10 m × 20 m region. The experiment began on April 1st, when the water temperature in the surrounding area was approximately 17 °C, and ended on October 31st of the same year, when the water temperature had dropped to approximately 17 °C after rising during the summer. During this time, when rainfall did not make up for evaporation losses from the mesocosms, the losses were replenished with distilled water.

### 2.2. Sampling and Measurements

Three water column samples were taken along the diameter of each tank, and mixed water samples from each mesocosm were collected using Plexiglas tubes (length = 1 m, diameter = 70 mm). A YSI portable equipment was used to detect electrical conductivity (EC), pH, and dissolved oxygen (DO) in real time. After filtration through Whatman GF/C and extraction through acetone, the amount of chl *a* in the water column (chl *a*-W) was measured by spectrophotometric analysis [35]. For planktonic bacterial sequencing, an additional 200 mL of water was filtered through a 0.22 μm membrane (Sartorius UK Limited, Surrey, UK) and replicated three times and then stored at −80 °C before DNA was extracted.

Additionally, sediment samples were collected and synchronized with tiny grabs to separate the surface (0–2 cm) layer into two sections. For physical and chemical analysis, one portion was kept at 4 °C, and the other was kept at −80 °C until DNA extraction. Eight sediment variables, including total nitrogen (TNs), total organic nitrogen (TONs), total carbon (TCs), total organic carbon (TOCs), total phosphorus (TPs), loss on ignition (LOI), and chlorophyll *a* (chl *a*-S) were determined after thawing, air drying, and sieving. TNs and TCs were measured using Thermo Flash 2000 Elemental Analyzer (Thermo Scientific, Bremen, Germany). TPs was measured according to [36]. LOI was measured after combustion at 550 °C. Fluorometric analysis of chl *a*-S was performed [37].

Fluxes of oxygen, total alkalinity (TA), inorganic nutrients (ammonium [NH_4_^+^] and nitrate [NO_3_^−^]), TN and TP were measured monthly in each mesocosm. In situ rates of sediment oxygen demand and nutrient exchange were determined using benthic chambers. Each chamber contained 1.0 L of water and had an area of 0.01 m^2^ with two symmetrical 5 cm diameter side holes, which were above the sediment surface. The upper edge of the side walls had rubber plugs for water exchange with the surroundings. The sampling chamber was equipped with a sampling tube inserted into the sediment of each intermediate chamber. The chambers were operated as follows: first the chambers were submerged and the side holes were opened for 30 min to exchange water from the chambers, after which the side holes were closed and water samples were taken with a 50 mL syringe for initial oxygen and nutrient analysis; then the chambers were incubated for 24 h and then samples were taken for final oxygen and nutrient analysis. Soluble nutrient flux samples were filtered through GF/C filters and frozen for later analysis. NH_4_^+^-N, NO_3_^−^-N, TN, and TP were measured according to [36]. By using potentiometric titration and Gran assessment, TA was manually calculated after the increase in [H^+^] after the second equivalence point [38]. The fluxes of oxygen and nutrients (mg m^−2^ d^−1^) were calculated from the concentration differences between the end and the beginning of incubation [22].

### 2.3. DNA Extraction, Sequencing, and Bioinformatics

Microbial amplicon sequencing of water and sediment Total DNA was extracted using the PowerWater DNA Isolation Kit (QIAGEN, Crawley, UK) according to the manufacturer’s instructions, and then tested for DNA purity and concentration, respectively. Under the conditions of 1% agar-gel electrophoresis at voltage of 5 V/cm for 20 min, DNA integrity was measured using an electrophoresis device. Utilizing ultrasonic crushing equipment, the DNA was divided into 400 bp-sized fragments. Using the high-throughput sequencing platform Illumina HiSeq (Illumina Inc., San Diego, CA, USA), qualified DNA was sequenced, and the resulting raw sequence data were used for further analysis. On the Illumina Miseq PE300 platform, paired-end sequencing was carried out using purified amplicons that were pooled in equimolar proportions (Majorbio Company in Shanghai, China).

The analysis of the raw data obtained by sequencing was mainly performed using the QIIME2 analysis suite [39]. Raw reads were quality-filtered using the DADA2 plugin of QIIME2 [40]. Features that appear in only one sample were filtered. The denoised reads were classified using the QIIME2′s plugin feature classifier classify-sklearn with the SILVA (138-99-nb) classification reference [41]. For the subsequent analysis, we used the SRS (scaling with rank subsampling) method to normalize the filtered feature counts for each sample [42]. Then, functional annotation of prokaryotic was assigned by the program FAPROTAX on the normalized feature table [43].

### 2.4. Statistics and Data Analysis

The results were tested for normality (Shapiro–Wilk test), transformed to log(n + 1) if necessary, and analyzed for the effects of temperature, nutrients, and time on the measured variables by repeated measures analysis of variance (ANOVA) using the date as a covariate. In each nutrient treatment, monthly data were analyzed using one way ANOVA with the fixed factor temperature (ambient, heated) to test the monthly effect of temperature on each measured variable in the different trophic state. Repeated measures ANOVA required homogeneity of data with covariance, and all data were examined for homogeneity of variance using box plots and residual plots as well as Cochran’s test. We estimated the effect size for each measurable variable of each sampling period in order to evaluate the extent of temperature effects in various nutrition level treatments. T Using an effect size, which incorporates the mean, standard deviation, and number of replicates of the control and treatment groups, the response of each treatment relative to the control was standardized, condensing the results of the various replicates into a single response value for the treatment group relative to the control [44]. The term “significance” in the text refers to *p*-values that are less than or equal to 0.1 because of the ecological relevance of this value. Statistical Package for the Social Science (SPSS) 17.0 was used for all analyses.

Using R (v 3.5.2), additional statistical tests and analyses were performed [45]. All heatmaps were performed with a “pheatmap” package [46]. The “psych” package in R and the software Gephi were used to evaluate potential co-occurrence links between microbial functional genes through network analysis based on random correlation matrices (Version 0.9, Bastian et al., 2009). Additionally, further Spearman rank correlations were performed between functional features using normalized counts and positive, significant correlations were visualized as network plots (r < 0.6, p.adj > 0.05). The impacts of nutrient loading, functional genes, and microbial communities on the C and N fluxes at the conclusion of the experiment under two temperature treatments (ambient and heated) were summarized using structural equation models (SEM). The R package lavaan (version 0.6-5) was used for SEM operations [47]. The standardized root mean squared residual (SRMR ≤ 0.08), the comparative fit index (CFI ≥ 0.95), and the χ^2^
*p* value (*p* > 0.05) were employed with typical significant criteria to evaluate the overall fit of the SEM [47,48]. Functional gene abundances and microbial abundances in this instance were spontaneously log-transformed to produce a normal distribution. By eliminating unimportant paths from an a priori model that encompassed all potential interactions, the model was chosen. We eliminated temperature interactions from the model because there were no detectable effects of direct ground temperature (either ambient or heated). These decisions led to the final model’s best fit.

## 3. Results

### 3.1. Experiment Water Temperature

The ambient treatment water temperature was 17 °C at the beginning of the experiment, reaching a maximum of 33.9 °C in July and a minimum of 13.9 °C in April. During the heating treatment, the water temperature was maintained at an average of 4.5 °C above the ambient temperature and the temperature difference was kept within a predetermined ± 0.3 °C (Appendix A).

### 3.2. Overall Analysis

The result of repeated measures analysis of variance (ANOVA) is shown on Table 1. Time significantly affected eight of nineteen variables indicating an up/down seasonal trend in changes of these affected variables. There were no interactions of temperature and nutrient level except chl *a*-W and flux of NO_3_^−^-N, but the latter was only marginally significant (*p* < 0.1). Nutrient influenced most of the variables, always by increasing values of benthic nutrient fluxes and decreasing organic carbon and nitrogen of sediment in a highly significant level (*p* < 0.01). In these repeated measures analysis of variance, temperature influenced fewer variables than nutrients. Temperature increased EC in both nutrient levels. However, due to the great impact of nutrients, the effect of temperature on the measured variables might be masked. Thus, the effect of temperature should be analyzed in different nutrient treatments, separately. In addition, previous studies revealed temperature has a great influence on water chemistry in special months [49]. Therefore, in order to obtain possible effects of microbial communities and their functions on fluxes, we selected months for microbial sequencing at the beginning of the experiment (April) and when the system changed drastically (July, August, September).

### 3.3. Chl a-W and chl a-S Changes

Changes of chl *a*-W throughout the experiment were much different between two nutrient levels (Appendix A). Apparently, high nutrient level always showed much higher phytoplankton biomass than low nutrient (Table 1) in corresponding months, however, it was shown that warming had a negative influence on the increase of phytoplankton biomass in high nutrient mesocosms from July. Chl *a*-W was much lower in heated mesocosms but increased significantly in unheated ones in summer. On the contrary, the opposite effect of warming on phytoplankton biomass was found in nutrient-unenriched water, and the chl *a*-W in heated mesocosms increased significantly in the summer. Monthly effect sizes of chl *a*-W showed the same result: the biggest impact caused by heating reached −3.44 and 5.71 in two nutrient levels, which suggested two extremely opposing effects on the increase of phytoplankton biomass in different nutrient levels in warm weather.

In both trophic levels, changes of chl *a*-S did not show any sensitivity to higher temperature (Appendix A). However, the overall chl *a*-S in all treatments reflected a tendency of decrease from spring to summer. Moreover, monthly values of chl *a*-S in lower nutrient treatment were slightly higher than the other nutrient level in the initial period in this study (Table 1).

### 3.4. EC, pH and DO Changes

The values of pH, EC, and DO in the bottom waters in various experimental conditions are presented in Appendix A. Warming significantly affected the values of EC in both nutrient levels during the experiment period. EC in heated treatment were significantly higher than unheated treatment. EC was also significantly affected by nutrient level (Table 1), and higher nutrient level systems presented higher EC. The highest EC in high nutrient level was 581.5 ± 15.7 µs/cm in May, by contrast, it was 402.6 ± 12.96 µs/cm in low nutrient level. Monthly mean pH of bottom waters in unheated mesocosms with high nutrient level ranged from 7.40 to 8.69 and it ranged from 7.34 to 8.04 in heated mesocosms. At the beginning of the experiment, no significant difference was found in high nutrient treatment. However, in August and September, there was a significant difference between the two different temperature treatments in high nutrient level. In low nutrient treatment, significant differences were found between heated and unheated mesocosms but no obvious evidence proved that warming can significantly affect pH in low nutrient level. This pattern was also mirrored in the monthly effect sizes of EC and pH. Most of the time, effect sizes of EC were significantly different from zero and the low nutrient treatment with elevated temperature got higher effect sizes than high nutrient in warmer months. The effect sizes of pH were not significantly different from zero except for two months in both nutrient levels, separately. The direction (sign) of the elevated temperature effect sizes were all in accord with general predictions from theories [50] (positive for accelerate effect, negative for inhibition), so the negative effect sizes of pH in high nutrient treatment demonstrated that an elevated temperature of 4.5 °C could inhibit pH from increasing in summer.

### 3.5. Main Characteristics of the Sediment

Both TNs and TONs were affected by nutrient (Table 1), but TNs was not affected by warmer temperature TNs in both nutrient mesocosms. For TONs, initially there was no difference between heated and unheated treatments in any nutrient level, however, in the middle of the experiment, temperature caused a significant difference that TONs in unheated treatment were much higher than heated in high nutrient mesocosms. At the end of the experiment in high nutrient level, TONs in heated mesocosms tended to become the same as unheated treatment. In general, most TONs values were not affected by temperature. The overall effect sizes of TNs in both nutrient levels and TONs in low nutrient treatment were not significantly different from zero. At high nutrient level, TONs caused effect sizes of −2.39 and −1.28 in July and August, respectively.

Higher temperature did not affect TCs and TOCs throughout the experiment. Therefore, overall monthly effect sizes of TCs and TOCs were not significantly higher than zero. At the beginning of the experiment, TCs and TOCs of high nutrient level were decreasing until mid-July and began to rise in August. No trend was found in the changes of TCs and TOCs in low nutrient level (Table 1).

Overall TPs and APA were not affected by heating in either high nutrient or low nutrient mesocosms. At the beginning of the experiment, TPs in low nutrient treatment declined rapidly and reached its lowest point in May, then began to rise in summer. A climbing trend of TPs was found in high nutrient mesocosms (Table 1). APA in all treatments did not show and trend significantly throughout the experiment. In addition, overall effect sizes of TPs and APA were not significantly different from zero.

Overall LOI in the low nutrient treatment of both thermal regimes were much higher than high nutrient (Table 1). Despite higher mean LOI in low nutrient treatment, no significant difference was found between two different temperature mesocosms in monthly LOI with low nutrient and it did not show any trend throughout the experiment (Table 1), either. For LOI in a high nutrient level, it decreased before June but began to increase in summer, it seemed that heating in a sense suppressed this rising process and caused effect sizes of −3.42, −0.97 in July and August, separately.

### 3.6. Variation of Benthic Fluxes

Fluxes of TA, on the other hand, were not affected by warmer temperature significantly (Appendix A). However, TA fluxes of low nutrient treatment were much higher than high nutrient in May, then became lower in summer. Additionally, an elevated temperature did not lead any nonzero effect size of TA fluxes throughout the study.

Overall monthly mean fluxes of NH_4_^+^-N in the experimental phase ranged from −0.062 to 0.714 mg m^−2^ h^−1^. Uptake of NH_4_^+^-N was rare (Appendix A; shown as a negative value), occurring only once in one treatment during this study. The rates of NH_4_^+^-N release from sediment in low nutrient mesocosms were significantly affected by higher temperature especially in warmer months. Warming obviously accelerated the release rates of NH_4_^+^-N in low nutrient treatment in warmer months (August–October). Both NO_3_^−^-N and TN fluxes were variable over the sampling periods. Warming did not significantly affect nitrogen nutrient exchange at the sediment-water interface in high nutrient treatment, but in low nutrient treatment, the rates of nitrogen nutrient release with warmer temperature are becoming much higher than ambient. The same pattern was also shown in the monthly effect sizes of NH_4_^+^-N, NO_3_^−^-N and TN fluxes. In low nutrient treatment, temperature elevation always caused a higher effect size on nitrogen nutrients release compared with high nutrient sediments in corresponding months, especially in warmer seasons.

Monthly mean fluxes of o-P in high nutrient mesocosms ranged from −0.421 to 1.075 mg m^−2^ h^−1^, in low nutrient it ranged from near zero (−0.017) to 0.153 mg m^−2^ h^−1^ (Appendix A). Uptake of o-P occurred only once (May) in the high nutrient level. The maximum release in ambient and heated treatments were 1.075 and 0.987 mg m^−2^ h^−1^ separately and these high rates occurred in late August. No significant difference was found between ambient and heated systems except October that the rate in unheated water was higher than heated. It did show that higher water temperature has a significant effect on monthly mean fluxes of o-P in low nutrient systems. The rates of o-P release from low nutrient level sediment were significantly increased by heating and the impact was much more severe compared to unheated treatment. In August and October, warming even changed the o-P exchange direction from uptake to release. The same pattern was found in TP fluxes in this study (Appendix A). Most fluxes of o-P in high nutrient systems were not affected by heating, however, dramatic change happened in low nutrient mesocosms that fluxes of TP were much increased by warming during warmer months (Appendix A, July to October). Monthly effect sizes of phosphorus release from sediment revealed a same phenomenon as the nitrogen nutrient dynamic-i.e., a higher effect size in low nutrient treatment with elevated temperature and a lower effect size in high nutrient level in warmer months.

During the experiment, most DO fluxes were not significantly affected by warming. Examination of the monthly pattern of benthic DO fluxes highlights two months: September and October, where DO fluxes in both nutrient treatments reached extreme values. Most DO fluxes were negative, indicating oxygen at the sediment-water interface was always consumed throughout the experiment. An effect size of 0.19 was led by heating in September in high nutrient level, another effect size of 1.57 was in April in low nutrient treatment.

### 3.7. Microbial Taxonomic Composition

The microbial community structure and composition of all samples were analyzed by sequencing data of 16S rRNA amplicons. Sequencing produced approximately the same number of clean reads, and rank subsampling was performed for normalization. The relative abundance of microbial communities in various experimental conditions were analyzed at phylum levels, and the community structure of microorganisms was composed of five dominant phyla (Proteobacteria, Bacteroidota, Actinobacteriota, Chloroflexi, and Cyanobacteria) (Appendix A). The microbial composition of water samples and sediment samples were different. Microbial community structure analysis based on average Bray–Curtis distances showed that most of bacterial communities of water samples clustered into one cluster, and those of sediment samples clustered into another cluster. The proportion of Bacteroidota, Actinobacteriota, and Cyanobacteria were higher (the means were 27.07%, 15.27%, and 11.74%, respectively) in the water samples than in the sediment samples (the means were 7.59%, 1.68%, and 0.46%, respectively). Seven water samples contained more than 50% abundance of Bacteroidota. The sediment samples contained less abundance of Cyanobacteria, but had higher abundance of Chloroflexi, Acidobacteriota, Desulfobacterota, and Nitrospirota.

### 3.8. Metabolism Potential of Microorganisms

For all samples of microorganisms including 23 functional C-cycle genes and 11 functional N-cycle genes in water and sediment, Figure 1 depicts the trends of relative abundance of C-cycle and N-cycle functional genes across time as well as TA and TN fluxes. The quantity and location of functional genes varied dramatically between ambient and heated conditions, as seen in Figure 1A’s heatmap. We observed that N cycle genes were significantly enhanced under heated conditions, both in the water column and in the substrate. While C-cycle genes showed a significant enhancement in the substrate under heated conditions. In the C cycle, in the water column, genes related to C degradation were enhanced in the LNT group of bacteria compared to the LN group, while in the substrate, genes related to C degradation were enhanced in both the LNT and HNT groups of bacteria compared to the LN and HN groups. This may be due to higher organic matter. In the N cycle, it is mainly the warming that enhances the nitrogen cycling processes in the water column and substrate, mainly the genes involved in nitrification processes and nitrogen fixation, especially in the heated high nutrient group of water column and substrate. Both anaerobic and denitrification-related functional genes were involved in the nitrogen removal process. The relative abundance of anaerobic-related genes and denitrification genes increased in the hot months (August and October). Denitrification-related functional genes peaked in October. The seasonal variation showed a dramatic effect of temperature and warming on the distribution of N-cycle genes.

### 3.9. Network Analysis

We built a network of co-occurrences between selected functional genes to examine potential connections between functional genes. Spearman’s correlation coefficient (R^2^ > 0.8, *p*-value < 0.01) indicated a significant positive correlation between functional genes (Figure 2), and the network consisted of dominant C- and N-related cycling genes. Compared to the LN group, the HN group showed a significant enhancement of the carbon degradation functional group, especially in the warmed substrate. Compared to the LNT group, the HNT group showed an enhancement of N cycling and C degradation functions. The results of the overall network analysis exhibited a clear enhancement of carbon degradation in the substrate by nutrients, while warming further enhanced N cycling in the water column and substrate.

### 3.10. Relationship between Carbon and Nitrogen Fluxes and Functional Genes of Microbial Carbon and Nitrogen Cycling

On samples of the water column and sediment, partial SEM tests were conducted to determine the factors influencing the variance of C and N fluxes in the heated and environmental groups. The impacts of nutrient loading, functional genes, and microbial communities on the C and N fluxes at the conclusion of the experiment, independently for the two temperature treatments (ambient and heated), were summarized using structural equation models (SEM). With a χ^2^ of *p* = 0.168, SRMR of 0.064, and CFI of 0.955, the overall SEM met each of the three model fit requirements.

SEMs results showed that the environmental factors of water and sediment (including nutrients and chl *a*) and the metabolism potential of microorganisms affected the carbon and nitrogen fluxes directly and indirectly, and the pathway varied among warming treatments (Figure 3). We found that temperature does not directly affect C and N fluxes at the water and sediment interfaces, but indirectly affects nutrient release from eutrophic waters by influencing environmental conditions in the water column and sediment, as well as microbial communities and functional genes. Environmental factors in the water column significantly enhanced C fluxes under heating conditions; meanwhile, environmental factors in the water column enhanced N cycling in the low nutrient group but inhibited N cycling in the high nutrient heating group under warming conditions. In addition, bacterial community and microbial functional structure are also important factors affecting C and N cycling. Warming alters the mode of action of microbial functions, and overall, these results suggest that warming enhances and alters genes associated with the C and N cycles.

## 4. Discussion

At the centennial scale, lake ecosystems need to face the dual dilemma of climate warming and water eutrophication [13,32]. Climate warming and eutrophication affect nutrient flux cycling at the water-sediment interface of lakes, which in turn impacts the ecological structure and function of lakes [6,33]. Our results demonstrate that a 4.5 °C increase in ambient temperature has different and complex effects on shallow water systems with different trophic status. Our experimental data suggest that in highly nutrient lakes, nutrient fluxes do not change significantly with warming, which is consistent with studies of fluxes at the shallow water-air interface [12], although increasing temperature is thought to be a driver of increase microbial activity, plant loss, and increased turbidity [51,52]. In addition, we found that in shallow mesotrophic clearwater ecosystems, the system will undergo severe changes with increasing temperature, which may be the cause of the transformation from mesotrophic shallow lakes to eutrophic systems as a result of global warming [20].

Our current mesocosm experiments suggest that climate warming will increase the risk of eutrophication in freshwater lakes, and that warming effects may be a combination of direct and indirect effects. Temperature variations may have a direct impact on the net mobilization of nutrients to pore water in mesotrophic shallow lakes, which will affect nutrient transport at the sediment-water interface [17,53,54]. Indirectly, because of the tight coupling between nutrient levels in phytoplankton and benthic algal communities [55,56], combined with increased activity of microbes that play a key role in the metabolic breakdown process [17]. Hence, warming may have an impact on the shift of primary production and autotrophic biomass from the benthic to the pelagic zone similarly to how eutrophication does [57,58]. In contrast, nutrient concentrations in the upper water are a more important regulator of fluxes in highly eutrophic shallow lakes than temperature, thus they might have an impact on diffusion gradients, which in turn could change the direction of fluxes into or out of the sediment.

Our study found that microbial activity mediated nutrient fluxes at the interface of water and sediment and responded differently to different nutrient level and warming treatments, affected diffusive gradients and thus the magnitude and direction of fluxes. This study looked into the distribution and quantity of community functional genes in sediments and water. Season has a significant impact on the quantity and distribution of functional genes, particularly those implicated in N cycles in the population. Among all the community functional genes investigated, the relative abundance of N cycle genes was highest in the HNT water and sediments, suggesting that the N cycle occurred more frequently in the heated group than in the environmental group and warming enhanced the microbial function of nitrate reduction and nitrate respiration as well as nitrogen fixation and nitrogen respiration. The relative abundance of C-cycle genes in the sediment of the heating group was also higher than that of the environmental group, and the sediment was mainly composed of functional groups related to carbon decomposition, such as chemoheterotrophy, aerobic chemoheterotrophy, and fermentation. According to the aforementioned findings, sediment nitrification took place more frequently in the heating group than in the environmental group, which is consistent with our earlier research [59]. Denitrification involves the critical process of nitrite reduction, and more functioning genes can encourage the removal of nitrogen from sediments. Understanding microbially mediated C and N cycles in eutrophic lakes requires evaluating the profiles of community functional genes and bacterial populations [23,25].

Our results suggest that although temperature had an impact on changes in carbon, nitrogen, and phosphorus concentrations in sediments over a number of months, the main determinant of nutrient concentrations in sediments was nutrient content, and these concentrations had their own seasonal patterns [60]. Previous studies have shown that benthic TA fluxes clearly increase with warming [61]. However, in the present study, the total benthic TA fluxes at both nutrient levels were not affected by the temperature increase. Perhaps this is because benthic TA fluxes are more influenced by precipitation (negative) and dissolution of carbonate-bearing minerals (positive) [62,63]. The carbon and nitrogen contents (LOI, TOCs, and TONS) in the sediments of the low nutrient treatment group were significantly higher than those of the high nutrient treatment group, indicating that the organic matter content in the sediments of the low nutrient group was higher. This might be due to the decomposition of macrophytes that died in the low nutrient medium. In our study, chl *a*-S showed a decreasing trend in all treatments, and this decreasing trend can be explained as a result of stripping and shading of planktonic microalgal mats [64]. Additionally, it’s crucial to remember several typical seasonal traits of shallow sediments in temperate environments, such the early spring establishment of benthic algae communities, vigorous photosynthesis, and low respiration rates [65]. Due to the increased nutrient needs of benthic microalgae and bacteria in the spring, shallow sediments frequently serve as a sink for inorganic and organic nutrients [61,66]. As more heterotrophic material is consumed and broken down by the action of microbial C-degrading enzymes (hydrolase and oxidase) and as new C inputs are reduced in the late summer and early fall, nutrient fluxes are directed out of the sediment [67,68].

Our findings demonstrate that heat greatly accelerates the release of nitrogen nutrients in the lower nutrient groups from summer sediments. One explanation could be because lower water temperatures can limit bacterial degradation of organic matter, so that NH4^+^-N release is inhibited in unheated treatments [66,69]. NH4^+^-N efflux was relatively high from August to October due to warmer weather and a more stable sediment regime than in spring, allowing microorganisms in the sediment to form more ammonium and consume more nitrite [52,69] and the development of reducing conditions. This could be the cause of the sediments’ high ammonium and phosphate concentrations as well as incredibly low nitrite levels. In contrast, monthly N fluxes at high trophic levels were not affected by the high temperature of 4.5 °C, but were significantly different from those at low trophic levels. The relative abundance of denitrification genes, were very high in heated samples. Since they may affect diffusion gradients and, in turn, the direction of fluxes into or out of the sediment, N concentrations in the overlying water column appear to be more significant in N fluxes in shallow lakes at high trophic levels than temperature [70]. On the other hand, because of the intricate interactions between the processes of ammonification, nitrification, and denitrification, dissolved oxygen in the bottom water and to a lesser extent temperature have a significant impact on nitrogen fluxes [70,71]. In our experiments, whereas NO^3^-flux was almost zero in the summer, the sediment consumed a high amount of dissolved oxygen in all treatments. However, a low abundance of the anammox gene was found, suggesting that the denitrification process was primarily responsible for removing ammonium, nitrate, and nitrite from the sediments of high-nutrient lakes. An earlier study found that denitrification produced more N^2^ than ammonia oxidation in Taihu Lake, another eutrophic lake [72]. The abundance of organic matter in the sediments increased the need for electron acceptors (nitrite and nitrate), which made the environment unfavorable for the development of anammox [66]. High quantities of denitrification genes in warm sample sediments may have a significant role in cyanobacterial blooms [66]. Thus, in addition to demonstrating how sensitive microbial activities are to temperature, our findings also hint at a potential suppression of nitrification in low dissolved oxygen environments [52,73].

Our findings are consistent with prior research, which found that throughout spring, phytoplankton biomass was consistently higher in high-nutrient systems than in low-nutrient systems, manifesting as a turbid water phase or a clear water column dominated by macrophytes [4]. However, as the experiment progressed, the effect of warming on phytoplankton biomass in the water column was extremely opposite in summer-warming led to a significant decrease in chl *a*-W in the high nutrient system and an increase in chl *a*-W in the low nutrient system. Similar to the shift in primary producers brought on by eutrophication in shallow lakes, warming may induce primary production to shift from the benthos to the water column. This may be due to the decrease in chl *a*-S and heating-induced release of nutrients from the sediment in the low nutrient system. [74,75]. Interestingly, in the present experiment, the mesotrophic system was not pushed into eutrophic state by warming, however, comparison of two different temperatures showed that warming greatly accelerated the eutrophication process. Why would phytoplankton biomass decrease due to warming at higher nutrient levels? One possible reason is the shading effect of phytoplankton. A combined reason is that warming and nutrients encourage phytoplankton, which are best suited to take advantage of the combined supply of high light and nutrients at the water surface [76]. Another reason is that very high temperatures (nearly 38 °C in summer) may inhibit the growth of some phytoplankton species when biomass is high, because the competitive advantage and disadvantage provided by increased temperatures have different effects on the species-i.e., the temperature effects observed in phytoplankton communities include changes in composition and reductions in cell size [77].

## 5. Conclusions

Our experimental results demonstrate that under the currently anticipated increase in global warming and eutrophication, the intensity of nutrients released from sediments into the overlying water increases, indicating that organic matter was more susceptible to decomposition by a greater range of bacteria, particularly at warmer temperatures. Although anthropogenic activities leading to increased eutrophication are the main factor, the additive effects of global warming should still not be underestimated. In particular, shallow lakes that are closely influenced by humans will greatly increase their internal nutrient pools under the combined effects of warming and hydrology, which will further disrupt the nutrient balance of these shallow lakes under warming conditions. Our study is dedicated to provide prospective recommendations for lake managementin the face of climate change.

## Figures and Tables

**Figure 1 microorganisms-11-00910-f001:**
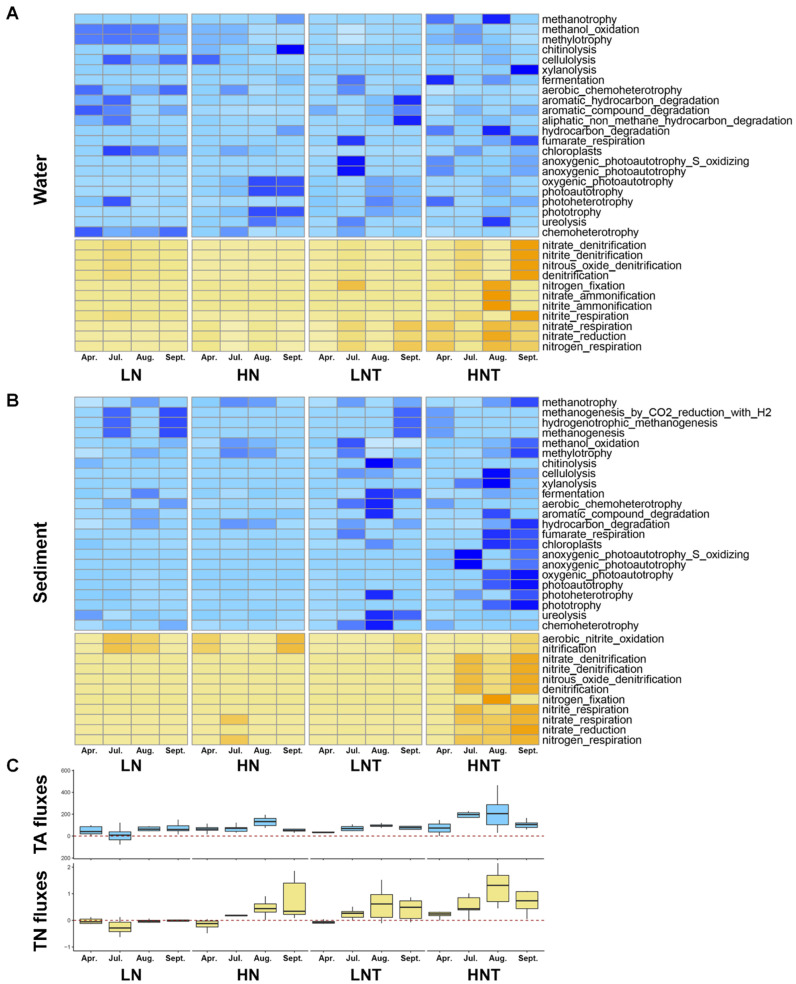
The relative abundances of C and N functional genes in the collected samples, as well as the corresponding fluxes of total alkalinity (TA) and total nitrogen (TN), changes between April and October in 12 mesocosms warmed by 4.5 °C above ambient and 12 unheated mesocosms. The relative abundance was calculated based on the percentage of functional gene to 16S rRNA gene of bacteria. (**A**) The water samples. (**B**) The sediment samples. (**C**) Fluxes of total alkalinity (TA) and total nitrogen (TN). The color shades in A and B indicate the amount of relative abundance, with dark colors indicating high and light colors indicating low. Blue for C functional genes and yellow for N functional genes.

**Figure 2 microorganisms-11-00910-f002:**
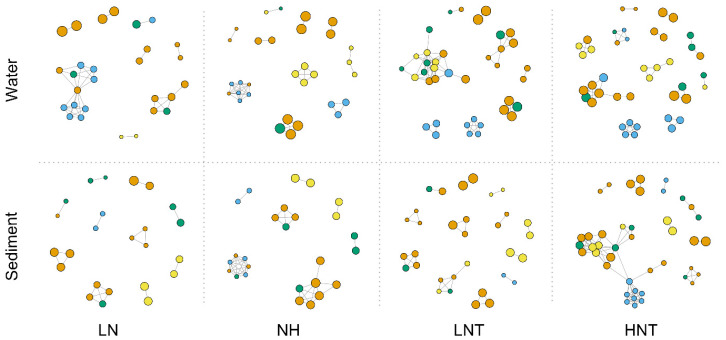
Network analysis between C and N functional genes in water and sediment samples. Each node represents a functional gene. Nodes were colored based on the functional gene (b) (R^2^ > 0.8 and *p* < 0.01). Blue for C functional genes and yellow for N functional genes. Node sizes weighted based on the node degree (the number of connections between a node and other nodes). Edges were colored based on the nodes they connected. Blue for C and yellow for N.

**Figure 3 microorganisms-11-00910-f003:**
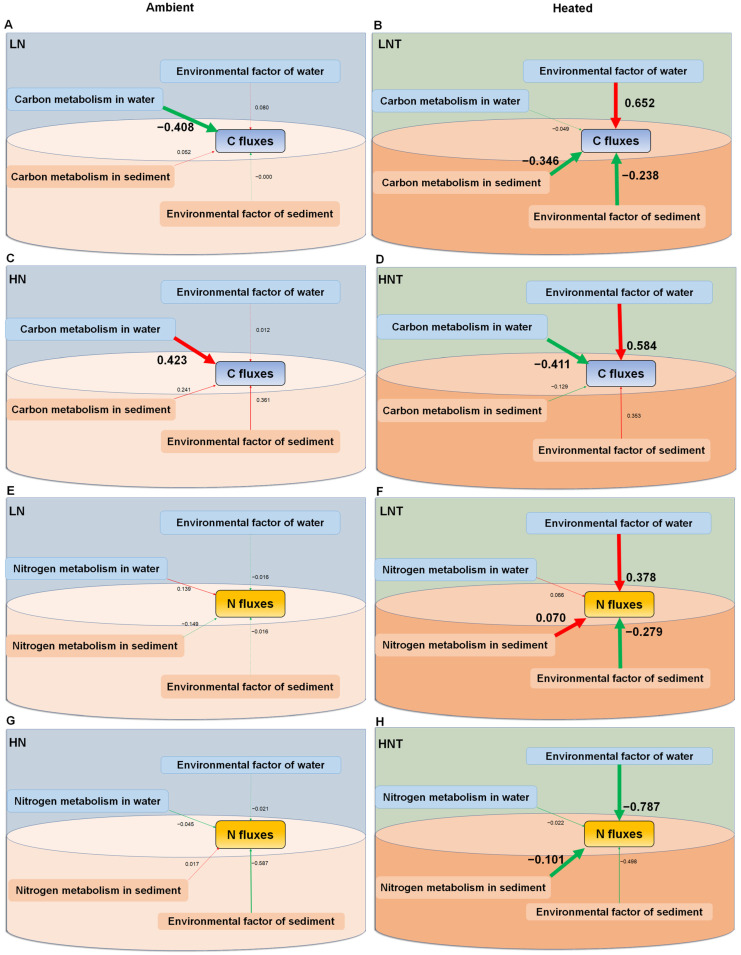
Structural equation models (SEM) showing the effects of nutrient loading and the environmental factor of water (TN_w_, TP_w_, and chl *a*-W) and sediment (TN_s_, TP_s_, and TOC) and the metabolism of microbial community (community diversity and functional gene), turbidity (Turb), and their interactions at the end of the experiment under ambient temperatures (**A**,**C**,**E**,**G**), continuous warming by +4.5 °C (**B**,**D**,**F**,**H**). Coefficients of determination (r^2^) are shown for all endogenous variables. Significant positive and negative effects, and insignificant interactions among variables are depicted in red and green arrows, respectively. Solid arrow thicknesses are proportional to the strength of the relationship. Numbers adjacent to arrows are standardized path coefficients and indicative of the effect of the relationship.

**Table 1 microorganisms-11-00910-t001:** Mean effects of temperature (T) and nutrient (N) on measured variables in 24 mesocosms between March and October (N = 42). Abbreviations used: ambient, A; heated, H; low nutrient, LN; high nutrient, HN; not significant, NS. Wherever results were significant, *p* values are indicated as ***: *p* < 0.01, **: *p* < 0.05, *: *p* < 0.1.

		Mean Value				Probability			
	Parameter	HN	HNT	LN	LNT	HN:HNT	LN:LNT	HN:LN	HNT:LNT
Water column	pH	7.93 ± 0.46	7.73 ± 0.27	7.94 ± 0.37	7.84 ± 0.29	*	NS	NS	NS
	DO (mg L^−1^)	3.56 ± 0.86	3.00 ± 0.57	3.99 ± 0.99	3.41 ± 0.71	NS	*	***	**
	Cond. (µs cm^−1^)	413.8 ± 76.4	470.5 ± 60.8	253.7 ± 80.7	320.0 ± 56.6	***	***	***	***
	TN (mg L^−1^)	0.86 ± 0.42	1.15 ± 0.52	0.08 ± 0.08	0.17 ± 0.17	*	**	***	***
	NH_4_^+^-N (mg L^−1^)	0.49 ± 0.41	0.53 ± 0.36	0.45 ± 0.33	0.53 ± 0.33	NS	**	***	**
	NO_3_^−^-N (mg L^−1^)	0.34 ± 0.37	0.38 ± 0.31	0.19 ± 0.25	0.21 ± 0.20	NS	NS	***	***
	TP (mg L^−1^)	1.32 ± 0.50	1.35 ± 0.41	0.13 ± 0.09	0.23 ± 0.16	NS	NS	***	***
	o-P (mg L^−1^)	0.86 ± 0.42	1.15 ± 0.52	0.08 ± 0.08	0.17 ± 0.17	***	NS	***	***
	TA (mg L^−1^)	84.2 ± 5.9	88.7 ± 9.2	44.4 ± 16.7	63.7 ± 9.56	NS	***	***	***
	SiO_3_^2−^ (mg L^−1^)	14.2 ± 7.12	17.8 ± 9.9	14.1 ± 9.5	19.1 ± 9.6	***	**	NS	NS
	Chl.*a* (mg L^−1^)	65.0 ± 59.7	27.1 ± 19.3	2.5 ± 1.1	10.5 ± 10.4	***	***	***	***

Sediment	LOI (%)	7.63 ± 0.81	7.59 ± 0.81	14.07 ± 0.64	13.43 ± 0.63	NS	NS	***	***
	TON (mg g^−1^)	2.79 ± 0.27	2.58 ± 0.33	4.98 ± 0.49	4.80 ± 0.32	**	NS	***	***
	TN (mg g^−1^)	2.90 ± 0.34	2.89 ± 0.34	5.16 ± 0.31	4.88 ± 0.19	NS	NS	***	***
	TOC (mg g^−1^)	27.17 ± 5.66	25.74 ± 5.15	49.58 ± 10.17	46.58 ± 11.42	NS	NS	***	***
	TC (mg g^−1^)	30.42 ± 3.92	30.89 ± 4.09	50.58 ± 3.72	47.40 ± 2.44	NS	NS	***	***
	TP (mg g^−1^)	1.55 ± 0.53	1.47 ± 0.56	0.17 ± 0.14	0.18 ± 0.32	NS	NS	***	***
	APA (mg g^−1^ h^−1^)	2.67 ± 0.34	2.64 ± 0.31	3.13 ± 0.36	2.98 ± 0.61	NS	NS	***	***
	Chl.*a* (mg g^−1^)	44.24 ± 7.68	42.58 ± 10.05	51.91 ± 15.72	47.45 ± 9.85	NS	NS	**	0.277

## Data Availability

Data available on request due to restrictions eg privacy or ethical. The data presented in this study are available on request from the corresponding author. The data are not publicly available due to this study is ongoing.

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
