# Peer review of "Climate Warming Does Not Override Eutrophication, but Facilitates Nutrient Release from Sediment and Motivates Eutrophic Process"

_microorganisms, 2023, doi:10.3390/microorganisms11040910_

Round 1
Reviewer 1 Report
The work is adequate and has merit for publication, since it presents relevant scientific information that will contribute to the development of research in the area of ​​climate change and its consequences for the functioning of ecosystems.
Author Response
Thank you for reviewing that help us improve our work.Reviewer 2 Report
This manuscript (microorganisms-2241853) suggests that climate warming promotes nutrient release from sediments and motivate eutrophic process, inducing a shift in microbial functional composition towards more conducive sediment carbon and nitrogen emissions. In particular, the resluts in this manuscript suggest that eutrophication process may be significantly accelerated in current projections of global warming, especially in shallow destratified clearwater lakes dominated by macrophytes. In conclusion, the authors argue that shallow lakes will greatly increase their internal nutrient pools under the combined effects of warming and hydrology, which will further disrupt the nutrient balance of these shallow lakes under warming conditions.
This manuscript has well-established hypotheses and adequate research methods to prove the hypothese, but still several issues seem insufficient to be published in the journal, Microorganisms.
1. Overall English editing seems necessary.
2. In materials and methods section, it should be described in more detail or using subdivided subheadings to make it easier for readers to understand.
3. The overall figures quality is not good enough. The figure legends need to be described in more detail (What do the colors in the figure represent? etc.)
4. Some or all of the supporting data should be included in the main manuscript to enhance understanding of this manuscript.
5. Figure s7 seems to need better modifing to make it easier to understand.
6. Is the legend of figure s7 correct? ex) The “s” represents water samples and “w” represents sediment samples.
Author Response
We really appreciate your thorough reviews and thoughtful comments that help us improving the quality of the revised manuscript. We have reorganized and rephrased the manuscript with more suitable descriptions of our results, especially in the materials and methods section. A more detailed explanation has been provided in the figures. Details are as follows. 1. We have revised the language. 2. As you suggested we have added subdivided subheadings. 3. We have re-edited the figure legends to meet the requirements for clear presentation. 4. Considering the volume of the text, we have added the supporting data in the Supplementary. 5. Revised. 6. Revised.Reviewer 3 Report
The study evaluated the effect of increasing temperature on eutrophication. The experiment was properly designed. The statistical analyses were correctly performed. The manuscript was well written. My comments are minimal.
1. It is recommended to use line numbers.
2. Please see specific comments in the pdf file.
3. Please see the marked text and check spelling throughout the manuscript.

Author Response
We express deep gratitude to your thorough reviews and thoughtful comments that improved the quality of the revised manuscript. Due to journal requirements, we are currently using the journal format and therefore have not added extra line numbers, and we hope we have your understanding. After double checking the manuscript, we corrected some typos and added the missing ones.Reviewer 4 Report
Shape comments
- Order the conclusions according to the hypotheses expected in the introduction
- In the sediment collection, mention the average depth of the lakes
- Change the word deep to depth, to express Depth
Content comments
They don't explain why they used aquarium heaters, because it's not the same as heat transfer from sunlight.
The heat transfer between the system and its surroundings is different through solar radiation than using passive heaters
Author Response
Thanks to your suggestion, we have re-edited the order of presentation in the conclusion section. Regarding the lake depth and sediment collection, we have described it in detail in the sampling section. And we verified that we artificially DEEP is a more appropriate description. Regarding the way of using heater to heat the experimental system, after our decade-long exploration, together with computerized intelligent real-time monitoring and circulating water system to homogenize the temperature, we tried to control it as close to natural level as possible. This method is also designed with reference to the exploration of other researchers and our experience. Relevant studies have been published for our heating system as follows. Zhang, H. , Zhang, P. , Wang, H. , Molinos, J. G. , Xu, J. . (2021). Synergistic effects of warming and eutrophication alert zooplankton predator-rey interactions along the benthic–pelagic interface. Global Change Biology. Zhang, P., Wang, T., Zhang, H., Wang, H., Hilt, S., Shi, P., Cheng, H., Feng, M., Pan, M., Guo, Y., Wang, K., Xu, X., Chen, J., Zhao, K., He, Y., Zhang, M., Xu, J., Heat waves rather than continuous warming exacerbate impacts of nutrient loading and herbicides on aquatic ecosystems. Environment International. 2022.168:107478 Wang, K.,Zhao, K.,Xiong, X.,Zhu, H.,Ao, H.,Ma, K.,Xie, Z.,Wu, C.,Wang, H.,Zhang, H.,Zhang, P.,Xu, J.Altered Energy Mobilization Within the Littoral Food Web in New Habitat Created by Climate-Induced Changes in Lake Water Level.Frontiers in Ecology and Evolution.2022 Shi, P.,Wang, H.,Feng, M.,Cheng, H.,Yang, Q.,Yan, Y.,Xu, J.,Zhang, M.Bacterial Metabolic Potential in Response to Climate Warming Alters the Decomposition Process of Aquatic Plant Litter-In Shallow Lake Mesocosms.Microorganisms.2022.10(7)
We express deep gratitude to the reviewers for your thorough reviews and thoughtful comments that improved the quality of the revised manuscript. We incorporated all of the reviewers’ suggestions and comments to make the modifications in the revised manuscript accordingly. And all the detailed comments are responded above specifically. Please refer to the revised manuscript.